# Impacts of Climate Change and Non-Point-Source Pollution on Water Quality and Algal Blooms in the Shoalhaven River Estuary, NSW, Australia

**Liu Wan [1,2,\*], Xiao Hua Wang [2] and William Peirson [3,4]**

[1] College of Oceanic and Atmospheric Sciences, Ocean University of China, Qingdao 266100, China
[2] The Sino-Australian Research Consortium for Coastal Management, School of Science, The University of New South Wales, Canberra, ACT 2601, Australia; x.h.wang@unsw.edu.au
[3] Water Research Laboratory, The University of New South Wales, Sydney, NSW 2052, Australia; b.peirson@newcollege.unsw.edu.au
[4] New College, The University of New South Wales, Sydney, NSW 2052, Australia
\* Correspondence: wanliu@stu.ouc.edu.cn

**Abstract:** This study quantifies some of the potential impacts of climate change and nutrient pollution to identify the most important factors on water quality changes and algal blooms in the study region. Three variables, air temperature and streamflow, representing climate change, and nutrient runoff, were varied in eight hypothetical scenarios to determine their impact on water quality and algal blooms by the calibrated and validated water quality model QUAL2K. Water quality was assessed by the concentrations of dissolved oxygen, total nitrogen, and phosphorus. Algal blooms were identified by phytoplankton concentration. An increase in air temperature of up to 2 °C resulted in an average increase of 3% in water temperature and 4.79% in phytoplankton concentration, and an average decrease of 0.48% in dissolved-oxygen concentration. Projected decreases in streamflow not only made the above phenomenon more significant but also significantly increased the concentration of total nitrogen, total phosphorus, and phytoplankton with the same pollution inputs. Under climate change, the biggest cause of concern for estuarine water quality is reduced streamflow due to decreases in rainfall. Water quality improvement is possible by regulating the concentration of non-point-source pollution discharge. By reducing nutrient runoff, the total nitrogen and phosphorus concentrations were also reduced, resulting in a significant increase in the dissolved oxygen concentration. This study highlights the most significant factors for managing water quality in estuaries subject to climate change.

**Keywords:** climate change; water quality; nutrient input; lower estuary; QUAL2K

## 1. Introduction

Coastal zones are among the most densely populated regions of the world because of their significant intrinsic value [1]. However, they are also among the most vulnerable ecosystems to the impacts of climate change [2,3]. Climate change affects freshwater resources, natural ecosystems, agriculture, forestry and fisheries, and other systems [4]. In the first decade of the 21st century, more than $1 billion was invested by governments in adaptation strategies to cope with water shortages in Australia [4]. Recent field measurements (2007–2019) have shown water temperatures in eastern Australian estuaries have increased by 2.16 °C (0.2 °C year$^{-1}$) and waters have become more acidic (0.09 pH units year$^{-1}$) [5]. The climate of coastal zones in southern Australia is predicted to become warmer and drier in general [5,6]. In the Shoalhaven catchment, although surface runoff in the far future (2060–2079) is likely to increase, multi-model predict average annual rainfall will decrease by 1.7% in the near future (2020–2039), along with a 9.7% decrease in surface runoff and 18% decrease in river discharge, which shows the strong drought tendency of this catchment [7].

Understanding the impacts of climate change on catchment runoff for estuaries is crucial for the reduction in property loss and better management of estuaries in the future. The main responses of coastal rivers to global climate change are warmer temperatures, changed runoff caused by altered rainfall patterns in the catchment, and regional variability in other climate variables [8]. To quantify the impacts of climate change on river water, the consequences of climate change can be divided into two aspects.

One of the primary effects of climate change is expected to be a rise in surface water temperatures [9,10] because of the close relationship between water and air temperatures [11]. Water temperature strongly influences the oxygen concentration, pH, and conductivity of the water, and hence the photosynthesis and respiration rates of estuarine microalgae and macrophytes [12]. A second climate-related response are changes in rainfall patterns in the catchment, thus affecting the freshwater flushing. Reduction in rainfall with climate change could increase the likelihood of freshwater flushing reduction, which will change water residence time, nutrient delivery, salinity, and phytoplankton growth [12,13]. There will also be changes in coastal geomorphology [14,15]. Reductions in streamflow can also lead to water quality impairment [16,17], and will further change both the demand for and supply of agricultural water, which is relevant to rural livelihoods and food security [18,19].

There have been recent impacts on river flows that can be attributed to climate change but there have also been direct impacts of human activities on coastal rivers over the past century [3,20]. Nutrients from the discharge of sewage have had a major direct impact on coastal waters [12]. Recognizing the effects of human-related pollution on water quality is one of the most important issues in effective water resource management [21,22]. For example, waste from agricultural livestock has been a long-term concern; current management practices do not adequately or effectively protect water resources from excessive nutrients, microbial pathogens, and drugs in the waste [23]. Excessive nutrients, especially nitrogen and phosphorus from urban and rural non-point sources, play an important role in river water quality and require management [24,25].

In addition to water quality, climate change and non-point source (NPS) pollution will affect algae growth. As one of the main potential threats to the water environment, algae could proliferate when favorable biological, chemical, and physical conditions are present [26]. Algal blooms, especially harmful algal blooms, threaten the survival of other species of the ecosystem indirectly and are also responsible for the degradation of the water quality in estuaries because of the severe biochemical oxygen demand (BOD) conditions [27]. In New South Wales (NSW), climate changes are expected to increase the frequency of harmful algal blooms due to changes in water temperature and catchment run-off [28]. Algal blooms in the Shoalhaven estuary (Figure 1) would threaten ecosystem services such as aquaculture and drinking water [29]. NSW takes pride in the long history of its oyster industry, which needs high-quality water in both the catchments and the ocean. Over the last few decades, the oyster industry has been competing with increased coastal development, with consequent water quality degradation by estuarine acidification, noxious or harmful algal blooms, and increased susceptibility of the oysters to disease outbreaks such as heat kill [28,30]. The results from the algal analysis could be useful to help inform the aquaculture industry [31].

To investigate the impacts of climate change and NPS pollution on river water quality quantitatively, a water quality model QUAL2K was chosen because of its universality and operability [32–35]. In 2012, the Environmental Protection Agency (EPA), Washington, DC, USA, published the current version of QUAL2K (Version 2.12). QUAL2K has been used for evaluating water quality in river catchments but can also be applied to predicting river water quality in response to climate change. QUAL2K is a one-dimensional steady-state hydraulic water quality model. As presently configured, an Excel workbook serves as the interface for QUAL2K: all inputs, outputs, and model execution are implemented within Excel. All interface functions are programmed in Excel's macro language Visual

Basic for Applications. All numerical calculations are implemented in Fortran 90 for speed of execution.

**Figure 1.** The Shoalhaven River downstream of Tallowa Dam following; Bomaderry Creek and Broughton Creek provide point sources (yellow points). The various estuary reaches (R1–R6) are delineated by the black triangles. Blue points with E labels: water sampling sites.

The aims of this study are: (1) to preliminarily understand Shoalhaven River water quality by analyzing monitoring data and setting appropriate water quality indicators, including water temperature, dissolved oxygen (DO), total nitrogen (TN), and total phosphorus (TP) concentration and (2) to investigate the water quality and algal bloom responses to climate change and NPS pollution scenarios by setting future scenarios in QUAL2K. This study will provide data support for the Shoalhaven River estuary and other similar estuarine water quality and algal bloom management under future climate change and nutrient input, which will be significant for adaptive management of estuarine areas.

## 2. Materials and Methods

### 2.1. Geography

The Shoalhaven River (Figure 1) is located about 150 km southwest of Sydney in eastern Australia and flows eastwards to the coast near Nowra. The river starts from the Great Dividing Range with an altitude of 1350 m and enters the sea at Crookhaven Heads as an open mature barrier estuary. The entire catchment is comprised of approximately 70% natural vegetation, 27% rural land, and 3% urban land [36]. It is approximately 300 km in length and has a total catchment area of 7300 km$^2$ [37]. The maximum and minimum monthly rainfalls are 117 mm in June and 42 mm in September, respectively [38]. The mean (2000–2019) monthly air temperature in the Shoalhaven River district ranges from a minimum of 6.6 °C in July to a maximum of 27.8 °C in January [38].

The area chosen for this study is the section of river downstream of Tallowa Dam (150°19′ E, 34°45′ S) which is from the tidal limit at the Burrier pumping station to Crookhaven Heads (Figure 1). One of the most populous regions in NSW, Nowra-Bomaderry has a population of over 30,000 [39]. The river plays an important role in daily life, providing fresh drinking water, agricultural and industrial water, and a vibrant river ecosystem. The water quality is clearly affected by residential, industrial, and agriculture wastewater discharge. Stormwater runoff from the major townships should also be taken into consideration. The lower-estuary floodplain is predominantly rural and used for dairy farming.

Some areas along the river do not have fencing to keep cattle from the river edge, which may result in further contamination.

### 2.2. QUAL2K Model Setting

The QUAL2K model permits simulations of water quality variables under climate change scenarios and pollution discharge scenarios in a river system during essentially steady flow periods [40,41]. The main river water quality variables relevant to climate change are water temperature and dissolved oxygen concentration, and these variables were examined in response to changes in air temperature and streamflow [40,42,43]. QUAL2K can also model how other water variables, such as phytoplankton concentration and the concentrations of nitrogen and phosphorus, respond to climate change. Other input variables in QUAL2K include dew point temperature, wind speed, cloud cover, and shade [44]; these are all kept at the QUAL2K constant values in this study.

For the study of river water quality and algal blooms with the QUAL2K model, the following were used [44]. The river system, from the headwater boundary to the downstream boundary, was conceptualized, with inputs and outputs along the river such as point sources (tributaries) and non-point sources. The conceptualized river system was then segmented into reaches comprised of equal length elements for model simulation, validation, and analysis. The reaches of the Shoalhaven River used in QUAL2K are shown schematically in Figure 2. Based on hydrology, geomorphology, and land use patterns, the Shoalhaven River downstream of the tidal limit was divided into two main parts: (1) the upper estuary between the Burrier pumping station and Nowra, which was subdivided into Reaches 1–3, and (2) the lower estuary between Nowra and the sea, which provided Reaches 4–6 [37] (Figure 1). In the upper estuary, sand is dominant in the channel bed due to tidal influences. The estuaries range in width from 50 m to 500 m, narrowing from the mouth going upstream. The point sources in Figure 2 are the tributaries of Bomaderry Creek and Broughton Creek (Figure 1).

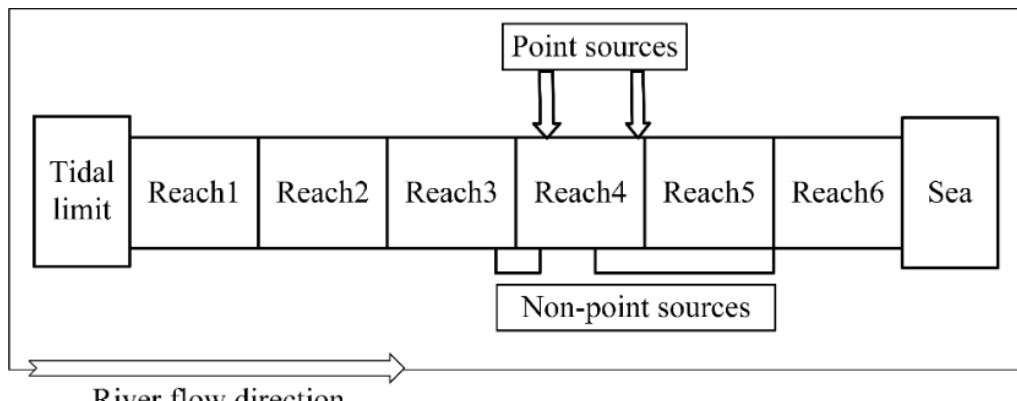

**Figure 2.** Segmentation of the Shoalhaven River downstream of Burrier into reaches for QUAL2K.

QUAL2K was used in this study to calculate water quality variables in the Shoalhaven River estuary. The model outputs of water temperature, DO concentration, DO saturation, the re-aeration coefficient ($k_a$), TN concentration, and TP concentration are shown in the result section. Full details of QUAL2K and all its variables are given in the QUAL2K documentation [44]. The key equations used by QUAL2K are given in Appendix A.

### 2.3. Data Collection

Data used for the model calibration and validation were collected in 2017 and 2018 [38,45–48], respectively.

### 2.3.1. Climate

The mean values recorded at the Nowra weather station in 2017 and 2018, were an annual average in air temperature (16.7 °C/16.6 °C in 2017/2018, respectively, the same below), dew-point temperature (10.6 °C/10.2 °C), wind speed (4.03 m/s/4.14 m/s), and cloud cover (71.2%/74%) [38,46] shown. Other than the air temperature, the mean values were applied to the model, which was used for the calibration and validation of the Shoalhaven model, respectively.

### 2.3.2. Inflows

The inflow at the tidal limit was taken to be the difference between the Tallowa Dam inflow and the Burrier pump station diversion. As the 2017 and 2018 data were not available, the average annual inflow from 2005 to 2009 at the tidal limit, 6.51 m$^3$/s, was used for both calibration and validation [47]. The 2017 and 2018 water quality inputs (water temperature, dissolved oxygen (DO), total nitrogen (TN), and total phosphorus (TP) concentrations) were obtained from the nearby monitoring site E-3 [48], located downstream of R2 (Figure 1). The average values from the 2005–2009 and 2017–2018 inflows are given in Table 1.

**Table 1.** Annual average water quality data for the Shoalhaven River inflow and point source inputs in 2017 and 2018.

| Variables | Shoalhaven River | Bomaderry Creek | Broughton Creek |
|---|---|---|---|
| | **2017** | | |
| Inflow (m$^3$/s) | 6.51 | 0.27 | 0.41 |
| Temperature (°C) | 16.4 | 19.5 | 19.6 |
| DO conc. (mg/L) | 7.00 | 6.88 | 4.48 |
| TN conc. (µg/L) | 350 | 400 | 425 |
| TP conc. (µg/L) | 8.00 | 66.7 | 650 |
| | **2018** | | |
| Inflow (m$^3$/s) | 6.51 | 0.02 | 0.19 |
| Temperature (°C) | 21.0 | 20.6 | 20.3 |
| DO conc. (mg/L) | 6.30 | 7.33 | 3.44 |
| TN conc. (µg/L) | 165 | 358 | 260 |
| TP conc. (µg/L) | 8.00 | 34.5 | 295 |

### 2.3.3. Point Sources: Bomaderry Creek and Broughton Creek

Bomaderry Creek and Broughton Creek are the main tributaries flowing into the lower Shoalhaven River estuary, downstream of Nowra [37]. The annual discharges of Bomaderry Creek and Broughton Creek in 2017 and 2018 [45] were set as point-source inputs to the Shoalhaven River (Table 1), together with the average annual water quality data from sampling sites E-5 and E-10 in Figure 1 [48].

### 2.3.4. Water Quality Measurements

The water quality variables (water temperature, and DO, TN, and TP concentrations) were monitored four times per year at monitoring sites [48], which were averaged into annual data used for calibration and validation of QUAL2K in this study. The annual average values of the water quality variables in 2017 and 2018 at the sampling sites (Figure 1) are shown in Table 2.

In Table 2, the values of water quality variables in 2017 and 2018 were different. In general, there was a higher water temperature and lower DO in 2018 than that in 2017. Moreover, the TN concentration in 2018 was lower than that of 2017, and the TP concentration in 2018 was higher in E-148 and E-294 than that in 2017. The values of water quality variables have inter-annual variation. Therefore, the study of hypothetical scenarios is essential to provide information in advance for future uncertain conditions.

**Table 2.** Annual average water quality indicator values from the Shoalhaven River monitoring sites in 2017/2018.

| Reach | Site Number | Temperature (°C) | DO Conc. (mg/L) | TN Conc. (μg/L) | TP Conc. (μg/L) |
|---|---|---|---|---|---|
| R2 | E-3 | 19.5/20.8 | 5.77/6.27 | 350/166 | 7.50/7.50 |
| R3 | E-414 | 19.7/20.5 | 5.28/3.90 | | |
| | E-342 | 19.7/20.6 | 5.33/4.02 | | |
| | E-148 | 19.6/20.5 | 4.83/3.56 | 350/305 | 22.5/52.8 |
| | E-149 | 20.7/20.4 | 4.86/3.53 | | |
| R4 | E-346 | 19.9/20.5 | 4.50/3.73 | | |
| | E-7 | 19.9/20.5 | 3.44/3.99 | | |
| | E-294 | 19.9/20.3 | 4.57/3.66 | 350/287 | 32.5/40.8 |
| R5 | E-295 | 20.4/20.6 | 4.26/3.7 | | |
| | E-548 | 14.5/19.9 | 5.00/3.93 | 450/352 | 56.3/41.5 |
| R6 | E-777 | 20.1/19.6 | 4.95/3.88 | | |
| | E-776 | 20.0/18.7 | 5.13/3.81 | | |

### 2.3.5. Non-Point Source Inputs

The collective pollution sources from Nowra to Terara were set as NPS in QUAL2K (the purple line in Figure 1). Due to a lack of NPS pollution monitoring, the difference in values between two adjacent water sampling sites (TN, between E-294 and E-548 in 2017; TP, between E-3 and E-148; E-148 and E-294; E-294 and E-548 in 2017) was taken as the NPS input reference in QUAL2K for the TN and TP concentration variables. The NPS input was set as a uniform discharge along the river, evenly distributed to each element in part of Reaches 3–4 and Reaches 4–5 (Figure 2) as the model boundary conditions.

### 2.3.6. Phytoplankton

Shoalhaven system's cyanobacteria were monitored as the concentration of Chlorophyll_a (μg/L), and shown in the annual water quality monitoring report of NSW water [49]. Cyanobacteria were regarded as the reference for phytoplankton in this study. The input phytoplankton concentration was based on 2017 data from monitoring site E-3 (Figure 1); this was set to 1 μg/L in the headwater. In the Shoalhaven River system, Chlorophyll_a exceeded the 5 μg/L threshold at most Shoalhaven system catchment sites at least once during the year [49]. Therefore, a phytoplankton concentration greater than 5 μg/L was taken to indicate an algal bloom [50]. Phytoplankton growth rates of 3.6/day were chosen, respectively, in three scenarios to study algal growth [51]. The other rates related to phytoplankton dynamics were taken from the QUAL2K manual [51,52].

### 2.4. QUAL2K Initial Conditions Setting

The initial condition setting of QUAL2K was mainly divided into three parts: catchment segmentation, input variable values, and system parameters.

In the catchment segmentation setting, the parameters of the Shoalhaven river reaching from the tidal limit to the estuary (Section 2.2) were specified in the Reach Sheet in QUAL2K. The Manning Formula was chosen as the hydraulic model [44].

In the input variable values setting, the values taken as a reference of initial conditions in the QUAL2K model, were obtained from data collection (Section 2.3). The Headwater Sheet contained the input values of annual inflow, water temperature, DO, TN, and TP concentrations (Table 1), and phytoplankton concentration (Section 2.3.6). The Point Sources Sheet contained the values of point sources (Table 1) from Bomaderry Creek and Broughton Creek. The values of air temperature, dew point temperature, wind speed, cloud cover (Section 2.3.1), and NPS (Section 2.3.5) were specified in the corresponding worksheets in QUAL2K separately, such as Air Temperature Sheet, Dew Point Temperature Sheet, and Wind Speed Sheet.

In system parameters setting, worksheets of parameters mainly contain Reach Rates Sheet, Rates Sheet, and Light-and-Heat Sheet [44]. The initial parameter values were adopted from QUAL2K documentation and adjusted during the calibration and validation process shown in Section 3.1 [44].

## 3. Results and Discussion

### 3.1. Calibration and Validation

In the present study, various parameter values were tried in the calibration and validation process. The performance of the QUAL2K model was analyzed using the correlation coefficient (R) and the root mean square error (RMSE) between monitored and simulated data, which were required to achieve the optimal solution as a whole. The model with the highest R and/or least RMSE was selected as the best model to describe the water quality variables in the Shoalhaven River. For example, solar shortwave and atmospheric longwave radiation are part of the air-water heat flux process [44]. During the calibration process, the Ryan–Stolzenbach method, for solar shortwave radiation with an atmospheric transmission coefficient of 0.87, and the Brutsaert method, the atmospheric longwave emissivity model for downwelling atmospheric longwave radiation, were chosen. This combination of methods gave the best fit of the QUAL2K water temperature to the observations (Figure 3).

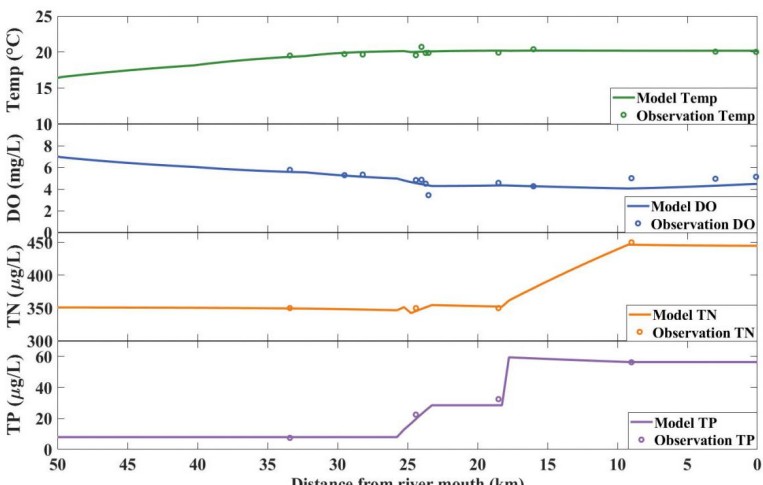

**Figure 3.** Model annual average water temperature and DO/TN/TP concentrations along the Shoalhaven River in 2017 compared with observations (model calibration).

Furthermore, the re-aeration coefficient $k_a$ (/day) in this model is closely related to the DO contributing to the biochemical processes. In calibrating the model, differences between the model and observed DO concentrations were reduced as much as possible by adjusting the re-aeration coefficients in each of the reaches. Calibration of the TN and TP concentrations was performed by adjusting the values of the NPS contaminant concentrations in the QUAL2K model with an NPS inflow assumption of 1.00 m³/s and water temperature assumption of 20 °C (Figure 3). The sharp increases in both the TN and TP concentrations in Figure 3 are because of the inflow of NPS pollution in part of Reaches 3–4 (TN concentration 510 µg/L; TP concentration 170 µg/L) and Reaches 4–5 (TN concentration 12,200 µg/L; TP concentration 30 µg/L). The calibration results in Figure 3 show an acceptable agreement between the annual average monitored data from 2017 and the model simulation.

Simulations were made with the relevant inputs for 2018 (Table 1) and compared with the observed values (Figure 4). An acceptable level of agreement was achieved.

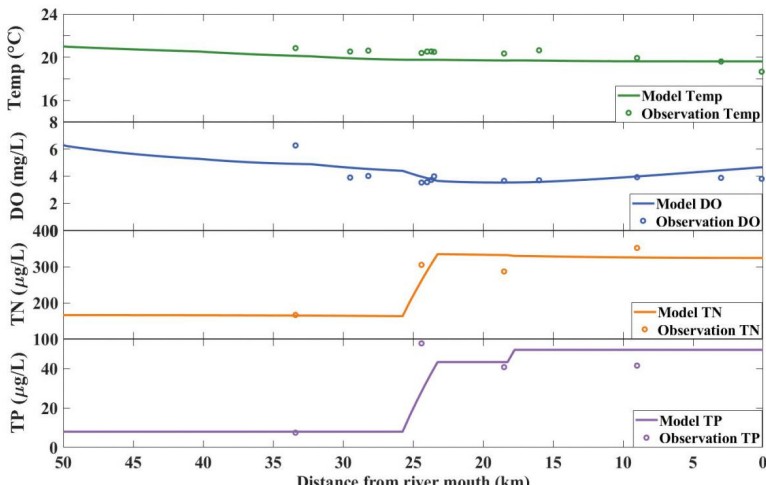

**Figure 4.** Model annual average water temperature and DO/TN/TP concentrations along the Shoalhaven River in 2018 compared with observations (model validation).

The performance measures for simulated temperature, DO, TN, and TP values for calibration and validation are listed in Table 3. DO, TN, and TP showed a strong correlation between the monitored and the simulated value in calibration and validation. The R of temperature in calibration and validation was 0.69 and 0.66, respectively. The RMSE was acceptable, with data evenly distributed on both sides of the simulation curve indicating a low bias in the model prediction (Figures 3 and 4). Therefore, the acceptable data model comparisons indicate the current QUAL2K's ability to successfully simulate the water quality of the Shoalhaven River.

**Table 3.** The QUAL2K model performance for the calibration and validation period.

| Model Performance Measurement Tools | Calibration (2017) | | | | Validation (2018) | | | |
|---|---|---|---|---|---|---|---|---|
| | Temperature | DO | TN | TP | Temperature | DO | TN | TP |
| Correlation coefficient (R) | 0.69 | 0.68 | 0.99 | 0.99 | 0.66 | 0.83 | 0.90 | 0.75 |
| Root mean square error (RMSE) | 0.25 | 0.49 | 3.91 | 2.71 | 0.69 | 0.76 | 31.17 | 11.96 |

*3.2. Future Scenarios Setting*

The 2017 results were used as a baseline because of its better model performance than that of 2018 (Table 3) for future changes study in air temperature, streamflow, and NPS pollution on water quality (water temperature, and DO, TN, and TP concentrations) and algal blooms.

The annual average temperature changes were predicted for the region up to 400 km inland of the coast in Australia for the years 2030, 2050, and 2070, relative to 1990 [4,53,54]. Climate change impacts on Australian water security, represented by the predicted changes in annual runoff (streamflow), were also predicted [55–57]. In the next fifty years, the air temperature in the regions of Australia up to 400 km from the coast will increase by 0.2 °C to 4.5 °C, and the streamflow in southeast Australia will decrease, in general, by up to 35%. The future climate change scenarios in this study are mainly based on climate change prediction.

The water quality response to climate change was assessed as follows. The mean air temperature (AT) was increased by 1.0 °C or 2.0 °C, and the streamflow (SF) was reduced by 35% from the 2017 values. Furthermore, the concentrations of the NPS pollutants were reduced by either 50% or 100%, representing a major reduction in human-related pollution [40]. These eight scenarios are summarized in Table 4, which also shows the along-river average percentage changes in the water quality variables of water temperature, and DO, TN, and TP concentrations.

**Table 4.** Scenarios investigated and percentage changes in water temperature (T), DO, TN, and TP concentrations, and phytoplankton concentration under the hypothetical scenarios for the total estuary (TE) and the lower estuary (LE).

| Hypothetical Scenarios | | | | Results in Percentage Changes | | | | |
|---|---|---|---|---|---|---|---|---|
| Scenario | ΔAT (°C) | ΔSF (%) | ΔNPS (%) | Range | ΔT | ΔDO | ΔTN | ΔTP | ΔPhy |
| 1 | 0 | 0 | 0 | TE | 0 | 0 | 0 | 0 | 0 |
| | | | | LE | 0 | 0 | 0 | 0 | 0 |
| 2 | +1.0 | 0 | 0 | TE | 1.5 | −0.24 | −0.03 | 0 | 2.34 |
| | | | | LE | 1.8 | −0.34 | −0.05 | 0 | 3.12 |
| 3 | +2.0 | 0 | 0 | TE | 3.0 | −0.48 | −0.07 | 0 | 4.79 |
| | | | | LE | 3.7 | −0.67 | −0.10 | 0 | 6.40 |
| 4 | 0 | −35 | 0 | TE | 0.87 | −2.8 | 2.9 | 25.5 | 15.90 |
| | | | | LE | −0.12 | −6.7 | 5.0 | 27.6 | 19 |
| 5 | +1.0 | −35 | 0 | TE | 2.5 | −3.1 | 2.8 | 25.5 | 19.41 |
| | | | | LE | 1.7 | −7.0 | 4.9 | 27.6 | 23.64 |
| 6 | +2.0 | −35 | 0 | TE | 4.1 | −3.4 | 2.8 | 25.5 | 23.11 |
| | | | | LE | 3.5 | −7.4 | 4.9 | 27.6 | 28.55 |
| 7 | 0 | 0 | −50 | TE | 0 | 1.5 | −8.9 | −18.7 | −0.23 |
| | | | | LE | 0 | 3.6 | −16 | −21.6 | 0.34 |
| 8 | 0 | 0 | −100 | TE | 0 | 3.1 | −18 | −37.4 | −0.45 |
| | | | | LE | 0 | 7.2 | −33 | −43.2 | 0.65 |

Notes: ΔAT is the change in air temperature; ΔSF is the change in streamflow; ΔNPS is the change in NPS; ΔT is the change in water temperature; ΔDO is the change in DO; ΔTN is the change in TN; and ΔPhy is the change in phytoplankton.

In this section, we also use QUAL2K to investigate the possibility of algal blooms in the Shoalhaven River estuary with changes in climate and NPS pollution. The same variables and the future scenarios of climate change and NPS pollution used previously (Table 4) were used to study changes in the algal bloom dynamics in the Shoalhaven River estuary.

For a more accurate analysis, a separate study of the water quality under the hypothetical scenarios was also conducted in just the lower estuary; the results are also shown in Table 4. In general, the changes in water quality and algal blooms were more obvious in the lower estuary than in the whole estuary.

The detailed changes in the water temperature, the DO, TN, and TP concentrations, and phytoplankton concentration in the different scenarios are described in the following sections.

### 3.3. Assessment of the Effects of Air-Temperature Increase Only

Figure 5 shows how the water temperature, the DO, TN, and TP concentrations, and phytoplankton concentration varied as the air temperature rose (scenarios 1–3 in Table 4). The (a) is the enlarged partial figure of DO concentration and (b) is the enlarged partial figure of phytoplankton concentration (same below in Figure 6).

When the air temperature was increased, the water temperature increased, whereas the DO concentration decreased, but only very slightly (enlarge as in Figure 5). There were no obvious changes in the TN and TP concentrations along the river in response to air temperature. The kinks in the graphs of the three concentrations are again because NPS pollutants were introduced into Reach 4 in the model (Figure 2). The DO concentration is one of the criteria for judging the quality of the water, which could negatively impact the sensitive species in water when it falls below 5 mg/L [58]. Note that this now pushes the DO concentration well below the minimum acceptable level of 5 mg/L.

The greater the increase in air temperature, the greater the increase in phytoplankton concentration. When the growth rate of phytoplankton was 3.6/day in QUAL2K, there

were algal blooms at the entrance to the ocean for air-temperature increases of 1 °C and 2 °C.

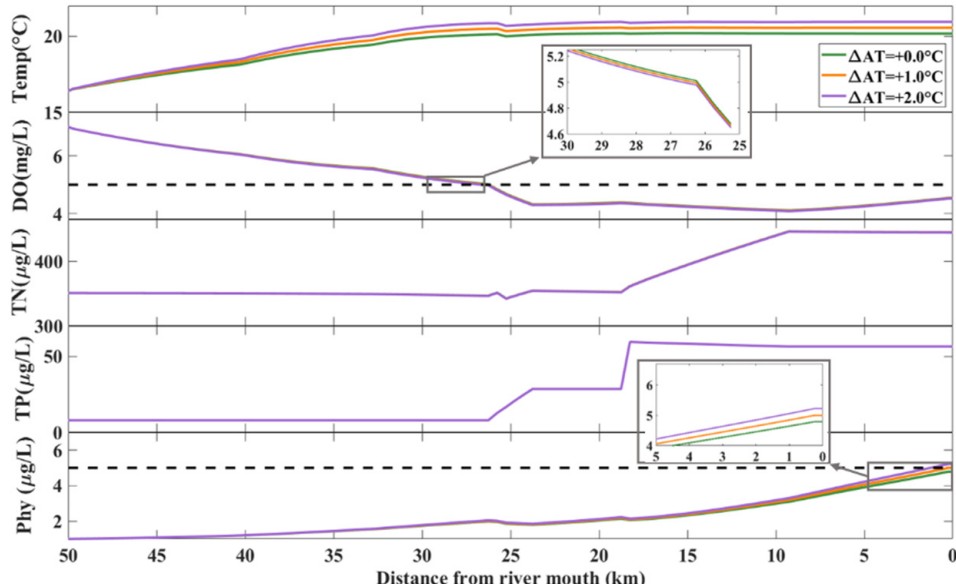

**Figure 5.** Water temperature/DO/TN/TP/phytoplankton concentrations along the Shoalhaven River with the incremental increases ΔAT in air temperature from 0 °C to 2 °C but under 2017 air temperature and streamflow conditions. The black dashed line in the DO concentration is the minimum acceptable level of 5 mg/L, and the black dashed line in the phytoplankton concentration is at 5 μg/L, which is the phytoplankton concentration threshold for an algal bloom (the same is below in Figures 6 and 7).

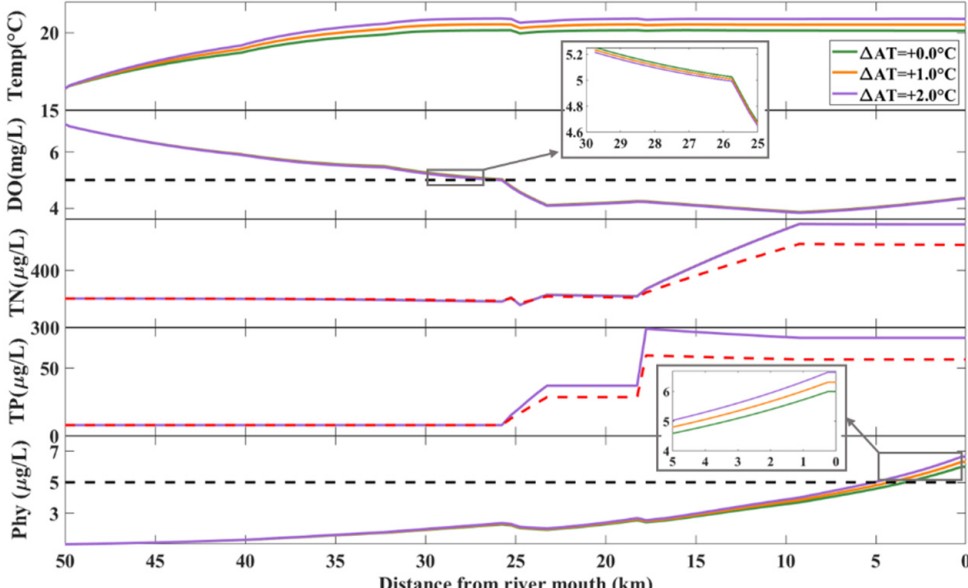

**Figure 6.** Water temperature/DO/TN/TP/phytoplankton concentration along the Shoalhaven River with the incremental increases ΔAT in air temperature from 0 °C to 2 °C with a decrease of 35% in the streamflow. The black dashed line in the DO concentration is the minimum acceptable level of 5 mg/L, and the black dashed line in the phytoplankton concentration is at 5 μg/L, which is the phytoplankton concentration threshold for an algal bloom. The red dashed lines in the TN and TP concentrations show the levels for no change in temperature or streamflow (from Figure 5).

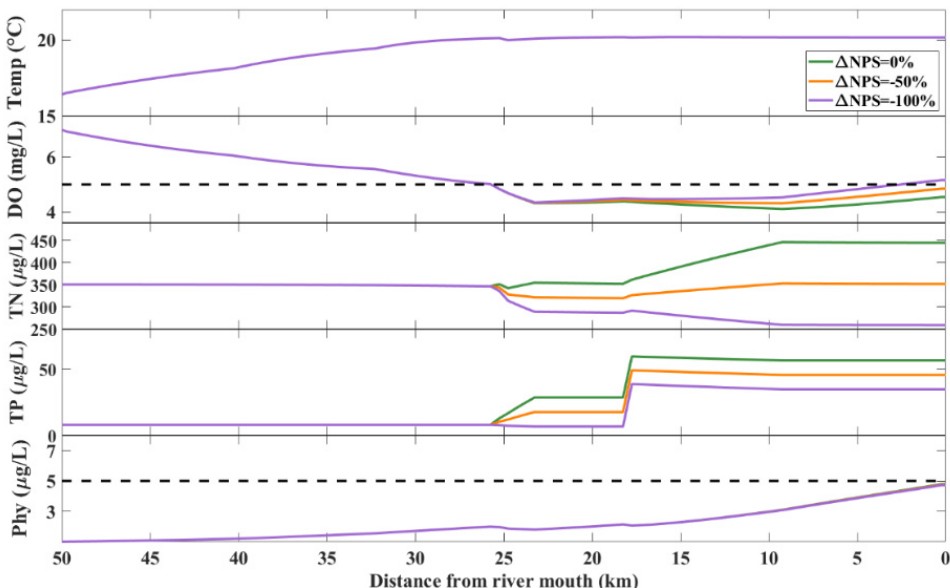

**Figure 7.** Water temperature/DO/TN/TP/phytoplankton concentration along the Shoalhaven River with the incremental changes (decreases) ΔNPS in NPS reduction from 0% to −100% but with 2017 air-temperature and streamflow. The black dashed line in the DO concentration is the minimum acceptable level of 5 mg/L, and the black dashed line in the phytoplankton concentration is at 5 µg/L, which is the phytoplankton concentration threshold for an algal bloom.

### 3.4. Assessment of the Effects of Air-Temperature Increase with Reduced Streamflow

Figure 6 shows the results of an air-temperature increase combined with reduced streamflow (scenarios 4–6 in Table 4). The water temperature again increased slightly but the TN and TP concentrations now showed an obvious increase with temperature in the lower estuary: the TN concentration increased by a maximum of 36 µg/L or 8.0%, and the TP concentration by 16 µg/L or 32% above the corresponding values with no change in air temperature or streamflow. Therefore, there was a greater effect on TP concentration than on TN concentration with a decreased streamflow.

More algal blooms were observed when the air-temperature increases were combined with a 35% decrease in streamflow (Figure 6). Decreased streamflow increases water residence time, which could increase the algae population [12,59]. With a 35% reduction in streamflow, the algae have about an extra 0.27 days because of the increase in the residence time (Equations (A12) and (A13)).

### 3.5. Assessment of the Effects of Changes in NPS Pollution Input

Changes in the water temperature and the DO, TN, and TP concentrations corresponding to changed nutrient inputs (scenarios 7 and 8 in Table 4) are shown in Figure 7. With a reduction in NPS pollution along the river, there were obvious decreases in the TN and TP concentrations in the lower estuary, by up to 187 µg/L or 42% and 21.9 µg/L or 77%, respectively.

There was very little difference in the phytoplankton concentrations when the NPS pollution was reduced. Clearly, the decrease in TN and TP concentrations due to the reduction in NPS pollution had no effect on algal blooms. Therefore, in this study, the concentration of nutrients was not a limiting factor for algal blooms in the Shoalhaven River estuary.

The DO concentration showed an obvious increase in the lower estuary with the decrease in NPS pollution because the decrease in TN and TP concentrations led to a decrease in DO consumption. The DO concentration increased by 0.60 mg/L at the entrance to the ocean when the NPS pollution was decreased by 100%, moving back above the

minimum acceptable level of 5 mg/L. Therefore, a decrease in NPS pollution would benefit the estuarine water quality.

## 4. Mitigation Option for Water Quality Improvement under Climate Change

As results show above, the increase in air temperature and reduction in streamflow in the Shoalhaven River under climate change could lead to water quality deterioration. The control of NPS pollution discharge, however, could benefit the water environment. In addition, overpopulation increases the possibility of environmental pollution [60]. With the local population set to grow rapidly over the next half-century and beyond, the future status of the region's waterways will depend on trade-offs between different uses [61]. Therefore, the water quality improvement under climate change related to NPS discharge was investigated. Based on climate change, three additional scenarios in ΔAT and ΔSF were set as 0 °C and −35%, +1 °C and −35%, and +2 °C and −35%, respectively (Table 5). Considering pollution discharge increase in the future, the changes in NPS pollution were set as 0%, ±30%, ±50%, and ±70%, respectively, in each scenario for comparison with the model results in 2017.

**Table 5.** Water quality improvement scenarios under climate change in the Shoalhaven River estuary by NPS discharge control.

| Scenario | ΔAT (°C) | ΔSF (%) | ΔNPS (%) | | | | | | |
|---|---|---|---|---|---|---|---|---|---|
| 1 | 0 | −35 | 0 | −30 | −50 | −70 | +30 | +50 | +70 |
| 2 | +1.0 | −35 | 0 | −30 | −50 | −70 | +30 | +50 | +70 |
| 3 | +2.0 | −35 | 0 | −30 | −50 | −70 | +30 | +50 | +70 |

The results of along-river average percentage changes in water quality variables were shown in Table 6. Accordingly, in the Shoalhaven River estuary, a 30% reduction in NPS pollution discharge could maintain the current level of the TN concentration in water quality improvement scenarios (Table 6). Lower NPS pollutant loads, represented by 50% pollution discharge reduction simulations, show that the water quality could maintain both the TN and TP concentration of 2017. NPS pollution reduction of more than 70% is required to improve the water quality measured by TN, TP, and DO concentration in 2017. The increase in NPS pollution under climate change lead to a worse water quality condition in DO (−6.2%), TN (19.6%), and TP (61.2%) concentration. The phytoplankton concentration results also showed a decrease along with the NPS decrease and an increase along with the NPS increase, respectively. These results are consistent with a recent study that showed different pollution discharges would influence water quality classification [41].

**Table 6.** Percentage changes in water temperature (T), DO, TN, and TP, and phytoplankton (Phy) concentrations under the water quality improvement scenarios for the Shoalhaven River estuary.

| | Scenario 1 | | | | | Scenario 2 | | | | | Scenario 3 | | | | |
|---|---|---|---|---|---|---|---|---|---|---|---|---|---|---|---|
| ΔNPS | ΔT | ΔDO | ΔTN | ΔTP | ΔPhy | ΔT | ΔDO | ΔTN | ΔTP | ΔPhy | ΔT | ΔDO | ΔTN | ΔTP | ΔPhy |
| 0 | 0.8 | −2.7 | 2.7 | 25.5 | 2.3 | 2.5 | −3 | 2.8 | 25.5 | 3.3 | 4.1 | −3.2 | 2.8 | 25.5 | 4.3 |
| −30 | 0.8 | −1.4 | −4.4 | 10.2 | 2.1 | 2.5 | −1.6 | −4.4 | 10.2 | 3.1 | 4.1 | −1.8 | −4.4 | 10.2 | 4.1 |
| −50 | 0.8 | −0.5 | −9.2 | 0 | 2.0 | 2.5 | −0.7 | −9.2 | 0 | 3.0 | 4.1 | −0.9 | −9.2 | 0 | 4.0 |
| −70 | 0.8 | 0.4 | −14 | −10.2 | 1.9 | 2.5 | 0.2 | −14 | −10.2 | 2.9 | 4.1 | 0 | −14 | −10.2 | 3.9 |
| +30 | 0.8 | −4.0 | 10.1 | 40.8 | 2.5 | 2.5 | −4.2 | 10.1 | 40.8 | 3.5 | 4.1 | −4.5 | 10.0 | 40.8 | 4.6 |
| +50 | 0.8 | −4.8 | 14.9 | 51 | 2.7 | 2.5 | −5.1 | 14.9 | 51 | 3.7 | 4.1 | −5.3 | 14.8 | 51.0 | 4.7 |
| +70 | 0.8 | −5.7 | 19.7 | 61.2 | 2.8 | 2.5 | −6.0 | 19.7 | 61.2 | 3.9 | 4.1 | −6.2 | 19.6 | 61.2 | 4.9 |

## 5. Conclusions

To understand how water quality and algal blooms would respond to climate change and NPS pollution in the Shoalhaven River estuary, simulations were conducted with air temperature, streamflow, and NPS pollution changed both separately and in combination. Climate simulations showed that streamflow reduction and temperature increase were

likely to occur [5,7]. Algal blooms were likely to be more severe with reduced streamflow. Reducing NPS may mitigate climate change. The significant factors affecting water quality in the Shoalhaven River were (from the strongest to the weakest): a change in streamflow (hypothetical scenarios 4–6); a change in NPS pollutants (hypothetical scenarios 7–8); and a change in air temperature (hypothetical scenarios 1–3).

Moreover, the water quality in the lower estuary was more adversely affected under these scenarios and thus should be considered a priority in water resource management in future environment planning to mitigate the effects of climate change (Table 3).

Concerning algal blooms, an air-temperature increase and/or a streamflow decrease could lead to an increase in phytoplankton concentration; a streamflow decrease had a greater effect because of the longer residence time of the phytoplankton. The larger the growth rate of the phytoplankton, the greater the possibility of algal blooms, especially in the lower estuary. When the growth rate was more than 3.6/day, algal blooms would occur when the air temperature was increased.

Our study highlights that changes in climate would bring negative impacts on river water quality and algal blooms. In addition, the modeling and evaluation results from water quality improvement scenarios indicate that NPS discharge control is a feasible measure to mitigate the deterioration of water quality caused by climate change.

Finally, a more comprehensive dataset of observations would improve the modeling of water quality. Additionally, during periods of low streamflow, saltwater intrusion caused by rises in tides and sea level may have a significant impact on river water quality and the ecology in estuaries. Therefore, further modeling studies including salinity and sea-level/tidal forcing at the ocean boundary would enhance our understanding of estuarine water quality and provide more comprehensive information to support the management of coastal rivers in response to climate change.

**Author Contributions:** Conceptualization, X.H.W. and W.P.; methodology, W.P.; software, L.W.; validation, L.W.; formal analysis, L.W.; data curation, L.W.; writing—original draft preparation, L.W.; writing—review and editing, X.H.W. and W.P.; supervision, X.H.W. and W.P. All authors have read and agreed to the published version of the manuscript.

**Funding:** This research received no external funding.

**Institutional Review Board Statement:** Not applicable.

**Informed Consent Statement:** Not applicable.

**Data Availability Statement:** Not applicable.

**Acknowledgments:** This is publication number No. 83 of the Sino-Australian Research Consortium for Coastal Management (previously the Sino-Australian Research Centre for Coastal Management). We would like to thank the Shoalhaven City Council for the access to the water quality and environmental flow data. This paper benefited from an editorial review by Peter McIntyre from UNSW Canberra. This research did not receive any specific grant from funding agencies in the public, commercial, or not-for-profit sectors.

**Conflicts of Interest:** The authors declare no conflict of interest.

## Appendix A

The following equations were used in QUAL2K [43].

The methods and formulations (Appendix) used in the validation were adapted from the calibration section.

*Appendix A.1. Water Temperature T [°C]*

$$\frac{dT_i}{dt} = \frac{Q_{i-1}}{V_i}T_{i-1} - \frac{Q_i}{V_i}T_i - \frac{Q_{out,i}}{V_i}T_i + \frac{E'_{i-1}}{V_i}(T_{i-1} - T_i) + \frac{E'_i}{V_i}(T_{i+1} - T_i) + \frac{W_{h,i}}{\rho_w C_{pw} V_i} + \frac{J_{a,i}}{\rho_w C_{pw} H_i} + \frac{J_{s,i}}{\rho_w C_{pw} H_i} \qquad (A1)$$

where $T_i$ is the temperature in element $i$, $t$ is the time [days], $Q_{i-1}$ is the inflow from the upstream element $i - 1$ [m$^3$/day], $V_i$ is the volume of element $i$, $Q_i$ is the outflow from element $i$ into the downstream element $i + 1$ [m$^3$/day], $Q_{out,i}$ is the total outflow from the element due to point and non-point withdrawals [m$^3$/day], $E'_i$ is the bulk dispersion coefficient between elements $i$ and $i + 1$ [m$^3$/day], $W_{h,i}$ is the net heat load from point and non-point sources into element $i$ [cal/day], $\rho_w$ is the density of water [g/cm$^3$], $C_{pw}$ is the specific heat of water [cal/g/°C], $J_{a,i}$ is the air-water heat flux [cal/cm$^2$/day)], $H_i$ is the depth of element $i$, and $J_{s,i}$ is the sediment-water heat flux [cal/cm$^2$/day)].

Appendix A.1.1. Air-Water Heat Flux: Solar Radiation

The atmospheric-attenuation model for solar shortwave radiation is [62]:

$$a_t = a_{tc}^{m(\frac{288 - 0.0065\text{elev}}{288})^{5.256}}$$ (A2)

$$m = \frac{1}{\sin \alpha + 0.15(\alpha_d + 3.885)^{-1.253}}$$ (A3)

where $a_t$ is the atmosphere attenuation, $a_{tc}$ is the atmospheric transmission coefficient (0.71–0.91), $m$ is the optical air mass, $\alpha$ is the sun's altitude in radians from the horizon, and $\alpha_d$ is the sun's altitude in degrees from the horizon $= \alpha \times (180°/\pi)$.

Appendix A.1.2. Air-Water Heat Flux: Atmospheric Long-Wave Radiation

The Brutsaert equation is used as the atmospheric longwave emissivity model in QUAL2K [63].

$$\varepsilon_{clear} = 1.24 \left( \frac{1.333224 e_{air}}{T_a} \right)^{1/7}$$ (A4)

where $\varepsilon_{clear}$ is the emissivity of longwave radiation from the sky with no clouds, $e_{air}$ is the air vapor pressure [mm Hg], and $T_a$ is the air temperature in °K. The vapor pressure from mm Hg to millibars is converted by a factor of 1.333224. The air vapor pressure [in mm Hg] is computed as:

$$e_{air} = 4.596 e^{\frac{17.27 T_d}{237.3 + T_d}}$$ (A5)

where $T_d$ is the dew-point temperature [°C].

*Appendix A.2.* DO $s_0$ Concentration per Day [mg/L/day]

$$s_0 = r_{oa} PhytoPhoto + r_{oa} \frac{BotAlgPhoto}{H} - r_{oc} FastCOxid - r_{on} NH4Nir - r_{oa} PhytoResp - r_{oa} \frac{BotAlgResp}{H} + OxRear$$ (A6)

where $r_{oa}$ is the ratio of oxygen to chlorophylla [g/mg], PhytoPhoto is the phytoplankton photosynthesis [µg/L/day], BotAlgPhoto is the bottom plant photosynthesis [µg/L/day], $r_{oc}$ is the ratio of oxygen to carbon oxidation $= 2.69 \frac{gO_2}{gC}$, FastCOxid is the fast CBOD (carbonaceous biochemical oxygen demand) oxidation [mg/L/day], $r_{on}$ is the ratio of oxygen to nitrification $= 4.57 \frac{gO_2}{gN}$, NH4Nir is the nitrification [mg/m$^3$/day], PhytoResp is the phytoplankton respiration [mg/m$^3$/day], BotAlgResp is the bottom plant respiration [mg/m$^2$/day], $H$ is the water depth (constant in a segment) [m], and OxRear is the oxygen re-aeration term, discussed below.

$$OxRear = k_a(T) \left( o_s(T, \text{ elev}) - s_0 \right)$$ (A7)

where $k_a$ is the re-aeration coefficient (/day), $o_s$ ($T$, 0) is the saturation concentration of dissolved oxygen in freshwater at 1 atm [mg/L], and $o_s$ ($T$, elev) is the saturation concentration of oxygen [mg/L] at temperature, $T$, and elevation above sea level, elev.

*Appendix A.3. Total Nitrogen (TN) Concentration [μg/L]*

$$TN = N_O + N_A + N_N + IP_N \tag{A8}$$

where $N_O$ is the nitrogen contribution from organic N, $N_A$ is from $NH_4$, $N_N$ is from $NO_2$ and $NO_3$, and $IP_N$ is from the phytoplankton intracellular nitrogen concentration.

*Appendix A.4. Total Phosphorus (TP) Concentration [μg/L]*

$$TP = P_O + P_I + IP_P \tag{A9}$$

where $P_O$ is the phosphorus contribution from organic P, $P_I$ is from inorganic P, and $IP_P$ is from the phytoplankton intracellular phosphorus concentration.

*Appendix A.5. Phytoplankton ($S_{ap}$) Concentration per Day [μg/L/day]*

$$S_{ap} = \text{PhytoPhoto} - \text{PhytoResp} - \text{PhytoDeath} - \text{PhytoSettl} \tag{A10}$$

where PhytoPhoto is the phytoplankton photosynthesis, PhytoResp is the phytoplankton respiration, PhytoDeath is the phytoplankton death, and PhytoSettl is the phytoplankton settling.

Phytoplankton photosynthesis is calculated as

$$\text{PhytoPhoto} = \mu_p \, a_p \tag{A11}$$

where $a_p$ is the phytoplankton concentration [μg/L] and $\mu_p$ is the phytoplankton photosynthesis rate [/day]. $\mu_p = k_{gp}(T) \, \phi_{Np} \, \phi_{Lp}$, where $k_{gp}(T)$ is the maximum photosynthesis rate at temperature $T$ [/day], $\phi_{Np}$ is the phytoplankton nutrient attenuation factor, and $\phi_{Lp}$ is the phytoplankton light attenuation coefficient.

*Appendix A.6. Residence Time [Day]*

The total residence time from the headwater to the downstream end of the *j*th element is shown below:

$$t_{i,j} = \sum_{k=1}^{j} \tau_k \tag{A12}$$

where $t_{i,j}$ is the travel time [day] and $\tau_k$ is the residence time in the *k*th element [day], given by

$$\tau_k = \frac{V_k}{Q_k} \tag{A13}$$

where $V_k = A_{c,k} \, \Delta x_k$ is the volume of the *k*th element [m$^3$], with $V_k$; $A_{c,k}$ its cross-sectional area [m$^2$], $\Delta x_k$ its length [m], and $Q_k$ the streamflow [m$^3$/s].

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
