# Peer review of "Impacts of Climate Change and Non-Point-Source Pollution on Water Quality and Algal Blooms in the Shoalhaven River Estuary, NSW, Australia"

_water, doi:10.3390/w14121914_

Round 1

Reviewer 2 Report

It is very hard to separate climate impacts from human ones on the changes in river water quality and discharge. Can you please better justify how did you do it?

Section 2.1: Please add more information about the rivers to the manuscript (e.g., annual discharge, average longitudinal slope of river, and a land-use map, if possible).

Section 2.2: QUAL2K is 1D steady-state model. Both river discharge and pollution loads are highly variable in time and space. How did you apply a steady-state model for such a case study?

Can you please add a separate section and explain about boundary/initial conditions?
